# A Decision Probability Transformation Method Based on the Neural Network

**DOI:** 10.3390/e24111638

**Published:** 2022-11-11

**Authors:** Junwei Li, Aoxiang Zhao, Huanyu Liu

**Affiliations:** School of Artificial Intelligence, Henan University, Zhengzhou 450046, China

**Keywords:** Dempster–Shafer evidence theory, average information content, interval information content, neural network, probabilistic information content

## Abstract

When the Dempster–Shafer evidence theory is applied to the field of information fusion, how to reasonably transform the basic probability assignment (BPA) into probability to improve decision-making efficiency has been a key challenge. To address this challenge, this paper proposes an efficient probability transformation method based on neural network to achieve the transformation from the BPA to the probabilistic decision. First, a neural network is constructed based on the BPA of propositions in the mass function. Next, the average information content and the interval information content are used to quantify the information contained in each proposition subset and combined to construct the weighting function with parameter *r*. Then, the BPA of the input layer and the bias units are allocated to the proposition subset in each hidden layer according to the weight factors until the probability of each single-element proposition with the variable is output. Finally, the parameter *r* and the optimal transform results are obtained under the premise of maximizing the probabilistic information content. The proposed method satisfies the consistency of the upper and lower boundaries of each proposition. Extensive examples and a practical application show that, compared with the other methods, the proposed method not only has higher applicability, but also has lower uncertainty regarding the transformation result information.

## 1. Introduction

Uncertain information [1] plays a significant role in many engineering applications, including multi-attribute decision-making [2,3,4], fault diagnosis [5,6,7], image processing [8,9], knowledge inference [10,11], risk assessment [12,13,14,15], and pattern classification [16,17]. Recent studies have focused on how to measure and handle information uncertainty, and a series of theories have been introduced, such as modeling information uncertainty based on the entropy function [18,19,20,21] and using Dempster–Shafer (DS) evidence theory [22,23,24,25,26,27,28,29,30] to deal with uncertain information. Compared with the traditional probability theory, the DS evidence theory can directly use the basic probability assignment (BPA) of multi-subset focal elements to express information uncertainty. However, a large number of elements in a proposition can make the credibility assignment too fragmented, thus causing certain complications in the decision-making process and reducing its precision. The BPA of multi-element propositions can respond to the support for each single-element proposition, which can be reasonably mapped to form probability in the frame of discernment [31,32,33,34,35,36,37].

Sudano [31] mapped the BPA of multi-element propositions to the probability of each single-element proposition according to a certain ratio using the belief and plausibility functions. Smets [32] equally distributed the BPA of multi-element propositions to each single-element proposition according to cardinality. However, this method is too conservative and does not fully use the known information. Pan [33] assigned the BPA of multi-element propositions according to the ordered weighted average operator (OWA) and then determined the final probability of each single-element proposition using the minimum entropy difference as a constraint. When a single-element proposition was not contained in any multi-element proposition, the allocation could still affect the single-element proposition, leading to anti-intuitive results. Deng [34] proposed a probability transformation method based on the belief interval, which first obtains the preference of each single-element proposition by the possibility degree, then quantifies the data information about the belief interval of each singleton based on the continuous interval argument ordered weighted average (C-OWA) operator, and finally calculates the support degree of the singleton based on quantitative data information to reasonably allocate the BPA of multi-subset focal elements. However, this method can result in extreme cases where the preference degree of a single-element proposition can be zero, which will further lead to a support degree of zero, thus meaning that the single-subset proposition is not assigned by the BPA to a multi-element proposition. Huang [35] proposed a probabilistic transformation method based on the Shapley value. This method requires that the degree to which each single-element proposition contributes to the multi-element propositions is determined; then, these are transformed according to the contribution degree. Since the marginal probability of a single-element proposition may be zero, which can mean that the proposition is not considered in the assignment of multi-element propositions, this may lead to inaccurate transformation results. Li [36] proposed a probability transformation method based on the ordered visibility graph (OVG). The OVG network was constructed according to the BPA order, and then the weight of each single-element proposition was obtained according to the proposition edges and cardinality. According to the weight, the BPA of multi-element propositions was transformed into the probability of each single-element proposition. This method only uses the out-degree and in-degree of the nodes to determine the weight and does not make full use of the influence of nodes with larger shadows. Chen [37] improved Li’s method as follows. First, the OVG network was constructed by each proposition order of information volume, and the weighted adjacency matrix was constructed with proposition edges or belief entropy. Then, the weight of each single-element proposition was obtained from the matrix and cardinality. Finally, the probability of each single-element proposition was obtained by assigning the BPA of multi-element propositions according to weight.

The key to the above methods is obtaining the distribution weight of the multi-element proposition. Inspired by these methods, under the premise of obtaining reasonable transformation results and minimizing the uncertainty of information, an efficient probability transformation method based on a neural network is proposed. The neural network is a computational model [38,39] designed to simulate the neural network of the human brain. Similar to human brain neurons, it consists of multiple nodes (neurons), which are interconnected to model the complex relationship between data. The connections between different nodes are given different weights, representing the influence of one node on another. Each node represents a specific function, which is calculated by combining the input and weight of other nodes. The result of the calculation is input into the activation function, which, in turn, provides the final output of the neuron and is passed on to the next neuron. The main questio is how to construct the neuronal network and changing weight, assigning the BPA of a proposition subset layer by layer according to weight until the probability of each single-element proposition containing the parameter is output. Then, the probabilistic information content (PIC) [31] is determined (with PIC being the dual form of Shannon entropy [40], which was first proposed by Sudano [31] and is widely used to measure the uncertainty of probability distribution) and, finally, optimal probability transformation results are obtained under the premise of PIC maximization. The main contributions of this study can be summarized as follows:(1).The BPA of a multi-element proposition with the largest cardinality is used as an input layer of a neural network, and the BPAs of the remaining existing propositions are used as bias units of the neural network. The hidden network layers are constructed according to decreasing order of cardinality for the proposition subset of the input layer and bias units. Finally, the probability of each single-element proposition is output by the output layer.(2).The interval information content (IIC) and average information content (AIC) are introduced to quantify the information contained in each proposition subset and combined to construct a weighting function containing the parameter. The weighting function reaches its extreme value at a certain point, in preparation for the use of the constraints below. After obtaining the PIC of the change based on the probability of each single-element proposition, the maximum PIC value is taken as a constraint to obtain the optimal transformation results.

The remainder of this paper is organized as follows. Section 2 briefly introduces the DS evidence theory and decision probability transformation methods. Section 3 describes the properties of belief entropy and PIC. Section 4 proposes a probability transformation method based on a neural network and proves its properties. Section 5 presents numerical simulation results to demonstrate the rationality and superiority of the proposed method compaed to the existing methods. A practical application of our proposed method is described in Section 6. Section 7 concludes the paper.

## 2. Preliminaries

In this section, preliminaries such as the DS evidence theory [24] and some existing probability transformation methods are briefly introduced.

### 2.1. DS Evidence Theory

Assume a set Θ is composed of *k* mutually exclusive elements, which can be expressed as: $\Theta =\left\{\theta_{1},\theta_{2},\ldots ,\theta_{k}\right\}$, where Θ is a frame of discernment; these elements are randomly combined to form the power set as follows:$$2^{\Theta }=\left\{\emptyset ,\theta_{1},\theta_{2},\ldots ,\theta_{k},\theta_{1}\cup \theta_{2},\cdots ,\theta_{1}\cup \theta_{2}\cup \theta_{3},\cdots ,\Theta \right\},$$
where ∅ denotes the empty set. If there is a function: $m:2^{\Theta }\to \left[0,1\right]$ satisfying the conditions of $m\left(\emptyset \right)=0$ and $\sum_{A\subseteq \Theta }m\left(A\right)=1$, then *m* is called the mass function, and $m\left(A\right)>0$ is the BPA of a proposition *A* .

If $Bel:2^{\Theta }\to \left[0,1\right]$ holds for $A\in 2^{\Theta }$, then:(1)$$Bel\left(A\right)=\sum_{B\subseteq A}m\left(B\right),$$
where $Bel\left(A\right)$ denotes the belief function, which represents the degree of trust that proposition *A* is true.

If $Pl:2^{\Theta }\to \left[0,1\right]$ holds for $A\in 2^{\Theta }$, then:(2)$$Pl\left(A\right)=\sum_{B\cap A=\emptyset }m\left(B\right),$$
where $Pl\left(A\right)$ denotes the plausibility function, which represents the degree of trust that proposition *A* is non-false.

Since the belief function Bel represents a degree of agreement with a proposition, and the plausibility function Pl indicates a degree of disagreement with the proposition; the interval formed by the belief and plausibility functions represents the uncertainty of the proposition. The intervals are formed by Bel and Pl, as shown in Figure 1.

### 2.2. Probability Transformation Methods

In the frame of discernment, $\Theta =\left\{\theta_{1},\theta_{2},\ldots ,\theta_{k}\right\}$, where $\theta_{i}\in \Theta$, when there is a multi-element proposition *D* in evidence, which has a large BPA, then the information assignment is too scattered, with large uncertainty, so it is difficult to make decisions directly based on the BPA. To facilitate the decision-making process and improve its accuracy, the following improvements have been introduced in recent studies.

Sudano [31] used Bel and Pl to obtain the probability of a single-element proposition by:(3)$$\mathrm{Pr}aPl\left(\theta_{i}\right)=Bel\left(\theta_{i}\right)+\varepsilon \cdot Pl\left(\theta_{i}\right),$$
where $\varepsilon =\frac{1-\sum_{\theta_{i}\in \Theta }Bel\left(\theta_{i}\right)}{\sum_{\theta_{i}\in \Theta }Pl\left(\theta_{i}\right)}$.

When only Pl is used, the probability of a single-element proposition is calculated by:(4)$$\mathrm{Pr}Pl\left(\theta_{i}\right)=\sum_{D\subseteq \Theta ,\theta_{i}\in D}\frac{Pl\left(\theta_{i}\right)}{\sum_{D_{i}\subseteq \Theta ,\left|D_{i}\right|=1,\cup_{i}D_{i}=D}Pl\left(D_{i}\right)}m\left(D\right).$$

In contrast, when only Bel is used, the probability of a single-element proposition is obtained by:(5)$$\mathrm{Pr}Bel\left(\theta_{i}\right)=\sum_{D\subseteq \Theta ,\theta_{i}\in D}\frac{Bel\left(\theta_{i}\right)}{\sum_{D_{i}\subseteq \Theta ,\left|D_{i}\right|=1,\cup_{i}D_{i}=D}Bel\left(D_{i}\right)}m\left(D\right).$$

Smets [32] provided an in-depth analysis of the reasonability of probability transformation in the decision-making domain and proposed the Pignistic probability transformation method, which is given by:(6)$$BetP\left(\theta_{i}\right)=\sum_{\theta_{i}\in D\subseteq \Theta }\frac{m\left(D\right)}{\left|D\right|},$$
where $\left|D\right|$ is the cardinality of a multi-element proposition *D* .

Pan [33] proposed a probability transformation method based on the OWA operator and entropy difference, and the average function can be expressed by:(7)$$M_{\left[Bel,Pl\right]}\left(\theta_{i}\right)=\frac{Bel\left(\theta_{i}\right)+Pl\left(\theta_{i}\right)}{2}.$$

The adopted normalization function is as follows:(8)$$\rho \left(\theta_{i}\right)=\frac{M_{\left[Bel,Pl\right]}\left(\theta_{i}\right)}{\sum_{\theta_{j}\in \Theta }M_{\left[Bel,Pl\right]}\left(\theta_{j}\right)}.$$

The probability of each single-element proposition obtained using the OWA operator is given by:(9)$$OWAP_{m}\left(\theta_{i}\right)={\left(T_{i}\right)}^{r}-{\left(T_{i-1}\right)}^{r},$$
where $T_{i}=\sum_{k=1}^{i}\rho \left(\theta_{k}\right)$ and $T_{0}=0$ . Since the probability of a single-element proposition contains an unknown variable *r*, $\min\left\{\left|E_{d}-H\right|\right\}$ is used to determine the variable *r*, where $E_{d}$ is the Deng entropy of the BPA, and *H* is the Shannon entropy of probability.

Deng [34] defined the belief interval and preference degree of a single-element proposition based on the belief and plausibility function, as follows:(10)$$l_{i}=Pl\left(\theta_{i}\right)-Bel\left(\theta_{i}\right),$$
(11)$$p_{ij}==\max\left\{1-\max\left(\frac{Pl\left(\theta_{i}\right)-Bel\left(\theta_{j}\right)}{l_{i}+l_{j}},0\right),0\right\},$$
(12)$$p\left(\theta_{i}\right)==\frac{1}{n-1}\sum_{j=1,j\neq i}^{k}p_{ij},$$
where i=1,2,…,k . The quantization of belief interval data was performed according to the C-OWA operator as follows:(13)$$\beta_{i}=\frac{Pl\left(\theta_{i}\right)+Bel\left(\theta_{i}\right)\cdot 2^{\left|\Theta \right|}}{2^{\left|\Theta \right|}+1}.$$

The preference degree was used to modify the quantized belief interval data to obtain the support degree of the single-element proposition as follows:(14)$$Sup\left(\theta_{i}\right)=\beta_{i}\cdot p\left(\theta_{i}\right).$$

The probability of a single-element proposition is given by:(15)$$ITP\left(\theta_{i}\right)=m\left(\theta_{i}\right)+\sum_{\theta_{i}\in B\subseteq \Theta }\varepsilon_{\theta_{i}}m\left(B\right),$$
where $\varepsilon_{\theta_{i}}=\frac{Sup\left(\theta_{i}\right)}{\sum_{j=1,\theta_{j}\in B\subseteq \Theta }^{\left|B\right|}Sup\left(\theta_{j}\right)}$.

Huang [35] proposed a probability transformation method based on the Shapley value, where the marginal probability of a proposition $\theta_{i}$ for a proposition *D* is expressed by:(16)$$MP\left(\theta_{i}\right)=m\left(D\right)-m\left(D\setminus \left\{\theta_{i}\right\}\right),$$
where $\theta_{i}\in D$, and D⊆Θ ; $D\setminus \left\{\theta_{i}\right\}$ denotes a subset of propositions *D* excluding $\theta_{i}$ . The average marginal probability contribution of $\theta_{i}$ in *D* is calculated by:(17)$$AMP_{D}\left(\theta_{i}\right)=\frac{1}{\left|D\right|!}\sum_{D\subseteq \Theta }\left[m\left(D\right)-m\left(D\setminus \left\{\theta_{i}\right\}\right)\right].$$

The probability of each single-element proposition is obtained by:(18)$$MPSV\left(\theta_{i}\right)=\sum_{D\subseteq \Theta ,\theta_{i}\in D}AMP_{D}\left(\theta_{i}\right).$$

After reordering each proposition according to the BPA from the largest to the smallest, Li [36] obtained a set of edges of each proposition based on the ordered visibility graph and obtained the weight of each single-element proposition according to the cardinality as follows:(19)$$g\left(\theta_{i}\right)=\frac{K\left(\theta_{i}\right)}{\sum_{D\in 2^{\Theta },\theta_{i}\in D}K\left(\theta_{i}\right)},$$
(20)$$K\left(\theta_{i}\right)=\sum_{\theta_{i}\in D,D\in 2^{\Theta }}\left(\frac{1}{\left|D\right|}K\left(D\right)\right).$$

The probability of a single-element proposition is calculated by:(21)$$OVGP_{m}\left(\theta_{i}\right)=\sum_{D\subseteq \Theta ,\theta_{i}\in D}\left(g\left(\theta_{i}\right)\frac{m\left(D\right)}{1-m\left(\emptyset \right)}\right),m\left(\emptyset \right)\neq 1.$$

Chen [37] ordered propositions according to the Deng entropy magnitude, and the Deng entropy of a proposition is calculated by:(22)$$IV\left(A_{i}\right)=-m\left(A_{i}\right)\log_{2}\frac{m\left(A_{i}\right)}{2^{\left|A_{i}\right|}-1}.$$

After ordering, which is denoted as $\left[\left(1,IV\left(A_{1}^{{}^{\prime }}\right)\right),\left(2,IV\left(A_{2}^{{}^{\prime }}\right)\right),\ldots ,\left(s,IV\left(A_{s}^{{}^{\prime }}\right)\right)\right]$, and where $A_{1}^{{}^{\prime }}$ is the ordered proposition, the network is constructed by the OVG. Then, the weighted adjacency matrix is obtained based on the set of edges with an internal element $b_{ij}$ . When there is a connection between $A_{i}^{{}^{\prime }}$ and $A_{j}^{{}^{\prime }}$, then $b_{ij}=1$; otherwise, $b_{ij}=0$ . The two edge weights are obtained using the node distance and belief entropy, respectively, as follows:(23)$$w_{ij}=\frac{b_{ij}}{\left|i-j\right|},$$
(24)$$w_{ij}=\frac{1}{2}\left(\frac{IV\left(A_{i}^{{}^{\prime }}\right)}{\sum_{a_{ki=1}}IV\left(A_{i}^{{}^{\prime }}\right)}+\frac{IV\left(A_{i}^{{}^{\prime }}\right)}{\sum_{a_{kj=1}}IV\left(A_{i}^{{}^{\prime }}\right)}\right).$$

Then, the degree of a focal element is calculated based on the edge weights as follows:(25)$$D\left(A_{i}^{{}^{\prime }}\right)=\sum_{j=1}^{s}w_{ij}.$$

The weight of a single-element proposition is calculated by:(26)$$w\left(\theta_{i}\right)=\sum_{\theta_{i}\in A{}^{\prime }}\frac{D\left(A^{{}^{\prime }}\right)}{\left|A^{{}^{\prime }}\right|}.$$

Finally, the probability of each single-element proposition is given by:(27)$$OVGWP\left(\theta_{i}\right)=m\left(\theta_{i}\right)+\sum_{\theta_{i}\in A\subseteq 2^{\Theta }}\frac{w\left(\theta_{i}\right)}{\sum_{\theta_{j}\in A}w\left(\theta_{j}\right)}m\left(A\right).$$

## 3. Shannon Entropy and Probabilistic Information Content

This section describes how to evaluate the performance of a transformation method after the probabilistic transformation is completed.

### 3.1. Shannon Entropy

Shannon [40] first introduced the concept of entropy into the field of information theory, defining the entropy of discrete finite sets. Assume a finite discrete set is defined as $F=\left\{f_{1},f_{2},\ldots ,f_{n}\right\}$, and its probability distribution is denoted by $P=\left\{p\left(f_{1}\right),p\left(f_{2}\right),\ldots ,p\left(f_{n}\right)\right\}$ . Then, the Shannon entropy of this set is obtained by:(28)$$H\left(F\right)=-\sum_{i=1}^{n}p\left(f_{i}\right)\log_{a}p\left(f_{i}\right),$$
where $\sum_{i=1}^{n}p\left(f_{i}\right)=1$; in this study, a=e, where *e* is Euler’s number. In a discrete set, when the probability of each element is equally distributed, i.e., $p\left(f_{1}\right)=p\left(f_{2}\right)=\ldots =p\left(f_{n}\right)=\frac{1}{n}$, then, the Shannon entropy is maximal, and it is given by $H_{\max}\left(F\right)=-\sum_{i=1}^{n}\frac{1}{n}\ln\frac{1}{n}$. The Shannon entropy is used to measure information uncertainty, and the greater information uncertainty is, the greater the entropy will be and vice versa.

### 3.2. Probabilistic Information Content

Sudano [31] developed a PIC method to evaluate transformation results. The PIC for the probability distribution $P=\left\{p\left(f_{1}\right),p\left(f_{2}\right),\ldots ,p\left(f_{n}\right)\right\}$ is calculated by:(29)$$PIC\left(F\right)=1+\frac{1}{H_{\max}\left(F\right)}\left(\sum_{i=1}^{n}p\left(f_{i}\right)\ln p\left(f_{i}\right)\right).$$

The PIC is expressed as a dual of the normalized Shannon entropy, and varies between zero and one. The smaller the PIC value is, the greater the information uncertainty will be, and vice versa. When PIC=1, there is no interference of uncertain information in decision-making, but when PIC=0, it will be impossible to make a decision based on the information. The PIC has often been used to evaluate the performance of probabilistic transformation methods.

## 4. Probability Transformation Based on the Neural Network

The existing methods assign the BPA of multi-element propositions to single-element propositions according to certain weights, but these methods have the defects of not fully considering the relationship between propositions and using the existing information, which will lead to inaccuracies in the weight factors generated in some special cases, and the probability transformation results that are obtained are counter-intuitive. To overcome the shortcomings of the existing methods, this study proposes a probability transformation method based on a neural network. In this section, the neural network construction and weighting function are introduced. The optimal transformation result is described in detail.

### 4.1. Neural Network Construction

A neural network has one input layer and one output layer, and the number of hidden layers and bias nodes depends on the actual situation of the evidence. In this paper, a neural network uses the ReLu function [41] as the activation function.

If the frame of discernment $\Theta =\left\{\theta_{1},\theta_{2},\ldots ,\theta_{k}\right\}\left(k\geq n\right)$ exists, the BPA $m\left(\theta_{1},\theta_{2},\ldots ,\theta_{n}\right)$ of the proposition $\left\{\theta_{1},\theta_{2},\ldots ,\theta_{n}\right\}$ with the maximum cardinality used as the input network layer. The BPAs of the remaining propositions are denoted as bias units, the propositions in the hidden layer are subsets of the input layer and the bias unit propositions, and the size of the cardinality decreases with the layer number. From the combination of elements in probability statistics, it is known that when the number of proposition subsets of the first hidden layer is $C_{n}^{n-1}=n$, the BPA of proposition subsets is expressed as $Nm\left(\theta_{1},\theta_{2},\ldots ,\theta_{n-1}\right)$, $Nm\left(\theta_{1},\theta_{2},\ldots ,\theta_{n-2},\theta_{n}\right)$, …, $Nm\left(\theta_{2},\theta_{3},\ldots ,\theta_{n}\right)$, where each proposition subset cardinality is (n−1 ). The number of proposition subsets of the second hidden layer is $C_{n}^{n-2}=\frac{n\times \left(n-1\right)}{2}$, and the number of proposition subsets of the *j*th hidden layer is $C_{n}^{n-j}=\frac{n\times \left(n-1\right)\times \ldots \times \left(n-j+1\right)}{j!}$, where $j!=j\times \left(j-1\right)\times \left(j-2\right)\times \ldots \times 2\times 1$. There are (n−2) hidden layers; the output layer gives the probability values of all single-element propositions in the frame of discernment as follows: $PNm\left(\theta_{1}\right)$, $PNm\left(\theta_{2}\right)$,…, $PNm\left(\theta_{k}\right)$. The neural network structure is presented in Figure 2.

Each neuron in the hidden layer is composed of multiple parts, and the initial value of a neuron is obtained based on the accumulation of weights and bias units and then activated using the activation function before obtaining the BPA of each focal proposition, as shown in Figure 3.

### 4.2. AIC and IIC Values

Assume the set of discernment $\Theta =\left\{\theta_{1},\theta_{2},\ldots ,\theta_{k}\right\}$, and a proposition $A_{i}\subseteq \Theta$ . In order to accurately quantify the information content of the proposition $A_{i}$, the average information content ( AIC ) and interval information content ( IIC ) based on belief and plausibility functions are defined as follows:(30)$$AIC\left(A_{i}\right)=e^{\left|2^{\Theta }\right|\left(\frac{Bel\left(A_{i}\right)+Pl\left(A_{i}\right)}{2}\right)},$$
(31)$$IIC\left(A_{i}\right)=e^{\left|2^{\Theta }\right|\left(Pl\left(A_{i}\right)-Bel\left(A_{i}\right)\right)}.$$

### 4.3. Weighting Function with Variable

This paper proposes a weighting function containing variable parameter *r*, which can obtain the weight of each proposition by combining the AIC and IIC. The weighting function is defined as follows:(32)$$W\left(A_{i}\right)=rAIC{\left(A_{i}\right)}^{1-r}+\left(1-r\right)IIC{\left(A_{i}\right)}^{r},$$
where 0≤r≤1.

**Theorem** **1**.
*The weight varies with a variable r; the function curve is neither monotonically increasing nor monotonically decreasing but has an extreme value point at a certain point.*


**Proof.** Let $W\left(A_{i}\right)=f\left(r\right)$, $AIC\left(A_{i}\right)=X$, and $IIC\left(A_{i}\right)=Y$ so that $f\left(r\right)=rX^{1-r}+\left(1-r\right)Y^{r}$, where 0<r<1, X≥1 and Y≥1, When r=0, then $f\left(0\right)=1$; when r=0, then $f\left(0\right)=1$ . The derivative of the weighting function is given by:
(33)$$f^{{}^{\prime }}\left(r\right)=X^{1-r}-rX^{1-r}\ln X-Y^{r}+\left(1-r\right)Y^{r}\ln Y,$$
since $f^{{}^{\prime }}\left(0\right)=X+\ln Y-1$ and $f^{{}^{\prime }}\left(1\right)=1-\ln X-Y$, regardless of the values of *X* and *Y*, $f^{{}^{\prime }}\left(0\right)$ and $f^{{}^{\prime }}\left(1\right)$ will always have opposite signs. According to the first sufficient condition for determining an extreme value, there exists $Z\in \left[0,1\right]$ so that $f^{{}^{\prime }}\left(Z\right)=0$; this point represents an extreme value of the function. □

If $A_{i}$ is a proposition of the *p*th hidden layer, assign its BPA to the subset proposition $a_{i}$ of the (p+1 )th layer as follows:(34)$$w\left(a_{i}\right)=\frac{W\left(a_{i}\right)}{\sum_{a_{j}\subseteq A_{i}}W\left(a_{j}\right)},$$
(35)Nm′ai=mai+∑ai⊆Ai⊆ΘAi=n−pwaiNmAi,
(36)$$Nm\left(a_{i}\right)=\max\left(0,Nm^{{}^{\prime }}\left(a_{i}\right)\right),$$
where $Nm\left(A_{i}\right)$ is the BPA of the proposition $A_{i}$ ; $m\left(a_{i}\right)$ is the bias unit; when there is no bias unit, then Nm′ai=∑ai⊆Ai⊆2ΘAi=n−pwaiNmAi . If $A_{i}$ is a proposition subset of the last hidden layer, its BPA is assigned to a single proposition subset $\theta_{i}$ as follows:(37)PNm′θi=mθi+∑θi∈Ai⊆ΘAi=2wθiNmAi,
(38)$$PNm\left(\theta_{i}\right)=\max\left(0,PNm^{{}^{\prime }}\left(\theta_{i}\right)\right).$$

At this point, the output probability contains variable parameter *r* .

**Theorem** **2**.
*According to the related literature [34], this transformation method can be justified by verifying the consistency of the upper and lower bounds: $Bel\left(\theta_{i}\right)\leq PNm\left(\theta_{i}\right)\leq Pl\left(\theta_{i}\right)$.*


**Proof.** Because PNmθi=mθi+∑θi∈Ai⊆ΘAi=2wθiNmAi, $PNm\left(\theta_{i}\right)\geq m\left(\theta_{i}\right)$, $m\left(\theta_{i}\right)=Bel\left(\theta_{i}\right)$, so $PNm\left(\theta_{i}\right)\geq Bel\left(\theta_{i}\right)$ holds; because $Nm\left(A_{i}\right)$ is jointly assigned by the BPA and bias nodes of the previous subset of propositions, so $Pl\left(\theta_{i}\right)\geq PNm\left(\theta_{i}\right)$; finally, inequality $Bel\left(\theta_{i}\right)\leq PNm\left(\theta_{i}\right)\leq Pl\left(\theta_{i}\right)$ holds. □

The above proof shows that the proposed method is a reasonable decision probability transformation method and, in the following, a specific example with large uncertainty is used to illustrate the calculation process of the proposed method.

**Example** **1**.
*Assume the frame of discernment is $\Theta =\left\{\theta_{1},\theta_{2},\theta_{3}\right\}$, and m is a mass function in Θ. Then, the corresponding BPA is given by:*

$$m\left(\theta_{1}\right)=0.2,m\left(\theta_{2}\right)=0.1,m\left(\theta_{1},\theta_{2}\right)=0.3,m\left(\theta_{2},\theta_{3}\right)=0.25,m\left(\theta_{1},\theta_{2},\theta_{3}\right)=0.15$$



First, a neural network is constructed based on the evidence, as follows: $m\left(\theta_{1},\theta_{2},\theta_{3}\right)$ is the input layer; $m\left(\theta_{1}\right)$, $m\left(\theta_{2}\right)$, $m\left(\theta_{1},\theta_{2}\right)$ and $m\left(\theta_{2},\theta_{3}\right)$ are the bias units; $\left\{\theta_{1},\theta_{2}\right\}$, $\left\{\theta_{1},\theta_{3}\right\}$, and $\left\{\theta_{2},\theta_{3}\right\}$ are the proposition subsets in the hidden layer.

Then, the AIC and IIC values of the hidden-layer propositions are obtained and used to construct the weight factors as follows:$$AIC\left(\theta_{1},\theta_{2}\right)=e^{\left|2^{\Theta }\right|\left(\frac{Bel\left(\theta_{1},\theta_{2}\right)+Pl\left(\theta_{1},\theta_{2}\right)}{2}\right)}=601.84,$$
$$IIC\left(\theta_{1},\theta_{2}\right)=e^{\left|2^{\Theta }\right|\left(Bel\left(\theta_{1},\theta_{2}\right)-Pl\left(\theta_{1},\theta_{2}\right)\right)}=24.53,$$
$$W\left(\theta_{1},\theta_{2}\right)=r\times 601.84^{1-r}+\left(1-r\right)\times 4.95^{r}.$$

Similarly,
$$W\left(\theta_{1},\theta_{3}\right)=r\times 81.45^{1-r}+\left(1-r\right)\times 270.43^{r},$$
$$W\left(\theta_{2},\theta_{3}\right)=r\times 403.43^{1-r}+\left(1-r\right)\times 601.85^{r}.$$

Then, assigning $m\left(\theta_{1},\theta_{2},\theta_{3}\right)$ to proposition subsets in the hidden layer according to the weights and combining the bias units, the BPA of each proposition in the hidden layer can be obtained by:$$\begin{array}{c} Nm^{{}^{\prime }}\left(\theta_{1},\theta_{2}\right)=m\left(\theta_{1},\theta_{2}\right)+\frac{W\left(\theta_{1},\theta_{2}\right)}{W\left(\theta_{1},\theta_{2}\right)+W\left(\theta_{1},\theta_{3}\right)+W\left(\theta_{2},\theta_{3}\right)}\times m\left(\theta_{1},\theta_{2},\theta_{3}\right)\\ =0.3+\frac{\left(r\times 601.85^{1-r}+\left(1-r\right)\times 4.95^{r}\right)\times 0.15}{r\times 601.85^{1-r}+\left(1-r\right)\times 4.95^{r}+r\times 81.45^{1-r}+\left(1-r\right)\times 16.44^{r}+r\times 99.48^{1-r}+\left(1-r\right)\times 6.05^{r}} \end{array}$$
$$Nm\left(\theta_{1},\theta_{2}\right)=\max\left(0,Nm^{{}^{\prime }}\left(\theta_{1},\theta_{2}\right)\right),$$
$$\begin{array}{c} Nm^{{}^{\prime }}\left(\theta_{1},\theta_{3}\right)=\frac{W\left(\theta_{1},\theta_{3}\right)}{W\left(\theta_{1},\theta_{2}\right)+W\left(\theta_{1},\theta_{3}\right)+W\left(\theta_{2},\theta_{3}\right)}\times m\left(\theta_{1},\theta_{2},\theta_{3}\right)\\ =\frac{\left(r\times 81.45^{1-r}+\left(1-r\right)\times 16.44^{r}\right)\times 0.15}{r\times 601.85^{1-r}+\left(1-r\right)\times 4.95^{r}+r\times 81.45^{1-r}+\left(1-r\right)\times 16.44^{r}+r\times 99.48^{1-r}+\left(1-r\right)\times 6.05^{r}} \end{array}$$
$$Nm\left(\theta_{1},\theta_{3}\right)=\max\left(0,Nm^{{}^{\prime }}\left(\theta_{1},\theta_{3}\right)\right),$$
$$\begin{array}{c} Nm^{{}^{\prime }}\left(\theta_{2},\theta_{3}\right)=m\left(\theta_{2},\theta_{3}\right)+\frac{W\left(\theta_{2},\theta_{3}\right)}{W\left(\theta_{1},\theta_{2}\right)+W\left(\theta_{1},\theta_{3}\right)+W\left(\theta_{2},\theta_{3}\right)}\times m\left(\theta_{1},\theta_{2},\theta_{3}\right)\\ =0.25+\frac{\left(r\times 99.48^{1-r}+\left(1-r\right)\times 6.05^{r}\right)\times 0.15}{r\times 601.85^{1-r}+\left(1-r\right)\times 4.95^{r}+r\times 81.45^{1-r}+\left(1-r\right)\times 16.44^{r}+r\times 99.48^{1-r}+\left(1-r\right)\times 6.05^{r}} \end{array}$$
$$Nm\left(\theta_{2},\theta_{3}\right)=\max\left(0,Nm^{{}^{\prime }}\left(\theta_{2},\theta_{3}\right)\right).$$

For simplicity, in the following, $Nm\left(\theta_{1},\theta_{2}\right)$, $Nm\left(\theta_{1},\theta_{3}\right)$, and $Nm\left(\theta_{2},\theta_{3}\right)$ are denoted by $g_{12}\left(r\right)$, $g_{13}\left(r\right)$, and $g_{23}\left(r\right)$, respectively. Next, the weight of each single-element proposition can be obtained by:$$W\left(\theta_{1}\right)=r\times {\left(e^{4\left(g_{12}\left(r\right)+g_{13}\left(r\right)+0.4\right)}\right)}^{1-r}+\left(1-r\right)\times {\left(e^{8\left(g_{12}\left(r\right)+g_{13}\left(r\right)\right)}\right)}^{1-r},$$
$$W\left(\theta_{2}\right)=r\times {\left(e^{4\left(g_{12}\left(r\right)+g_{23}\left(r\right)+0.2\right)}\right)}^{1-r}+\left(1-r\right)\times {\left(e^{8\left(g_{12}\left(r\right)+g_{23}\left(r\right)\right)}\right)}^{r},$$
$$W\left(\theta_{3}\right)=r\times {\left(e^{4\left(g_{13}\left(r\right)+g_{23}\left(r\right)\right)}\right)}^{1-r}+\left(1-r\right)\times {\left(e^{8\left(g_{13}\left(r\right)+g_{23}\left(r\right)\right)}\right)}^{r}.$$

Similarly, $W\left(\theta_{1}\right)$, $W\left(\theta_{2}\right)$, and $W\left(\theta_{3}\right)$ are denoted by $q_{1}\left(r\right)$, $q_{2}\left(r\right)$, and $q_{3}\left(r\right)$, respectively. The bias units are combined to obtain the probability of a single-element proposition by:$$\begin{array}{c} PNm^{{}^{\prime }}\left(\theta_{1}\right)=0.2+\frac{W\left(\theta_{1}\right)}{W\left(\theta_{1}\right)+W\left(\theta_{2}\right)}\times Nm\left(\theta_{1},\theta_{2}\right)+\frac{W\left(\theta_{1}\right)}{W\left(\theta_{1}\right)+W\left(\theta_{3}\right)}\times Nm\left(\theta_{1},\theta_{3}\right)\\ =0.2+\frac{q_{1}\left(r\right)}{q_{1}\left(r\right)+q_{2}\left(r\right)}\times g_{12}\left(r\right)+\frac{q_{1}\left(r\right)}{q_{1}\left(r\right)+q_{3}\left(r\right)}\times g_{13}\left(r\right) \end{array}$$
$$PNm\left(\theta_{1}\right)=\max\left(0,PNm^{{}^{\prime }}\left(\theta_{1}\right)\right),$$
$$\begin{array}{c} PNm^{{}^{\prime }}\left(\theta_{2}\right)=0.1+\frac{W\left(\theta_{2}\right)}{W\left(\theta_{1}\right)+W\left(\theta_{2}\right)}\times Nm\left(\theta_{1},\theta_{2}\right)+\frac{W\left(\theta_{2}\right)}{W\left(\theta_{2}\right)+W\left(\theta_{3}\right)}\times Nm\left(\theta_{2},\theta_{3}\right)\\ =0.2+\frac{q_{2}\left(r\right)}{q_{1}\left(r\right)+q_{2}\left(r\right)}\times g_{12}\left(r\right)+\frac{q_{2}\left(r\right)}{q_{2}\left(r\right)+q_{3}\left(r\right)}\times g_{23}\left(r\right) \end{array}$$
$$PNm\left(\theta_{2}\right)=\max\left(0,PNm^{{}^{\prime }}\left(\theta_{2}\right)\right),$$
$$\begin{array}{c} PNm^{{}^{\prime }}\left(\theta_{3}\right)=\frac{W\left(\theta_{3}\right)}{W\left(\theta_{1}\right)+W\left(\theta_{3}\right)}\times Nm\left(\theta_{1},\theta_{3}\right)+\frac{W\left(\theta_{3}\right)}{W\left(\theta_{2}\right)+W\left(\theta_{3}\right)}\times Nm\left(\theta_{2},\theta_{3}\right)\\ =\frac{q_{3}\left(r\right)}{q_{1}\left(r\right)+q_{3}\left(r\right)}\times g_{13}\left(r\right)+\frac{q_{3}\left(r\right)}{q_{2}\left(r\right)+q_{3}\left(r\right)}\times g_{23}\left(r\right) \end{array}$$
$$PNm\left(\theta_{3}\right)=\max\left(0,PNm^{{}^{\prime }}\left(\theta_{3}\right)\right).$$

The changing trends of the probability and the PIC value of single-element propositions with *r* are shown in Figure 4.

### 4.4. Optimal Probability Calculation of Single-Element Propositions

The larger the PIC value is, the lower the information uncertainty in the decision-making, the better the transformation results, and the more favorable the decision-making. Additionally, the higher the information uncertainty, the worse the transformation results and the less favorable the decision-making. According to [33], to obtain optimal transformation results, it is necessary to determine parameter *r* by maximizing the PIC as follows:(39)$$\begin{array}{c} \underset{r}{\arg max}\left(PIC\left(PNm\left(\Theta \right)\right)\right)\\ s.t\begin{array}{c} PIC\left(PNm\left(\Theta \right)\right)=1+\frac{1}{H_{\max}\left(PNm\left(\Theta \right)\right)}\left(\sum_{i=1}^{k}PNm\left(\theta_{i}\right)\ln PNm\left(\theta_{i}\right)\right) \end{array} \end{array}$$
where $H_{\max}\left(PNm\left(\Theta \right)\right)=-\sum_{i=1}^{k}\frac{1}{k}\ln\frac{1}{k}$. For Example 1, it holds that $max\left(PIC\left(PNm\left(\Theta \right)\right)\right)=0.2230$, r=0.91, so $PNm\left(\theta_{1}\right)=0.2755$, $PNm\left(\theta_{2}\right)=0.6380$, and $PNm\left(\theta_{3}\right)=0.0865$ . This transformation process is illustrated in Figure 5.

Next, a brief overview of the proposed method steps is given. The exact content of each step varies with the actual situation, but the main operations of each of the steps are as follows:

Step 1: Different propositions are combined to construct a neural network model;

Step 2: The AIC and IIC of proposition are obtained by Equations (Equation 30) and (Equation 31), weights are initiated combining the AIC and IIC by Equation (Equation 32), and the BPA of each proposition is assigned according to the weights until the single-element proposition probability containing the variable parameter *r* is output by Equations (Equation 34)–(Equation 38);

Step 3: The parameter *r* is determined according to the constraints and the optimal transformation results are obtained by Equation (Equation 39).

## 5. Analysis and Numerical Examples

In this section, a few examples are given to compare the transformation results of the proposed method with those of the other methods [31,32,33,34,35,36,37] to verify the rationality and accuracy of the method proposed in this paper. The method performances are evaluated based on the PIC value.

**Example** **2**.
*The frame of discernment is $\Theta =\left\{A,B,C\right\}$, and m is a mass function in Θ ; then, the BPA is given by:*

$$m\left(A\right)=0.2,m\left(B,C\right)=0.8$$



The probability transformation results of different methods for Example 2 are shown in Table 1.

In the probability transformation, since there is no information on single-element proposition *A* in the multi-element proposition, the assignment should be independent of the proposition *A*, and it is impossible to discern the difference between propositions *B* and *C* based on the known conditions. Intuitively, the BPA of a multi-element proposition should equally be assigned to the single-element propositions *B* and *C*. However, the PeBel transformation method cannot be used to obtain the transformation results, and the OWA method is influenced by the single-element proposition *A* when assigning the BPA of the multi-element proposition, which leads to counter-intuitive transformation results. The other probability transformation methods distribute the BPA of multi-element propositions equally according to the elemental cardinality to obtain a single-element proposition probability, which is consistent with the intuitive results.

**Example** **3**.
*The frame of discernment is $\Theta =\left\{A,B\right\}$, and m is a mass function in Θ; then, the BPA is given by:*

$$m\left(A\right)=0.5,m\left(A,B\right)=0.5$$



Step 1: According to the cardinality of propositions, $m\left(A,B\right)$ is used as the input layer, $m\left(A\right)$ denoted the bias unit, and the probability of a single-element proposition is obtained by the output layer.

Step 2: The AIC and IIC of the single-element proposition *A* and *B* are obtained by Equations (Equation 30) and (Equation 31), respectively, as follows:$$AIC\left(A\right)=20.0855,$$
$$AIC\left(B\right)=2.7183,$$
$$IIC\left(A\right)=7.3891,$$
$$IIC\left(B\right)=7.3891.$$

The weights of a single-element are calculated by Equation (Equation 32) as follows:$$W\left(A\right)=r\times 20.0855^{1-r}+\left(1-r\right)\times 7.3891^{r},$$
$$W\left(B\right)=r\times 2.7183^{1-r}+\left(1-r\right)\times 7.3891^{r}.$$

The probabilities of a single-element proposition are obtained by Equations (Equation 34), (Equation 37), and (Equation 38):$$PNm^{{}^{\prime }}\left(A\right)=0.5+\frac{W\left(A\right)}{W\left(A\right)+W\left(B\right)}\times 0.5$$
$$PNm\left(A\right)=\max\left(0,PNm^{{}^{\prime }}\left(A\right)\right),$$
$$PNm^{{}^{\prime }}\left(B\right)=\frac{W\left(B\right)}{W\left(A\right)+W\left(B\right)}\times 0.5$$
$$PNm\left(B\right)=\max\left(0,PNm^{{}^{\prime }}\left(B\right)\right).$$

The changing trends of the probability and PIC value with *r* are shown in Figure 6.

Step 3: Obtain the optimal probability by Equation (Equation 39) as follows:r=0.19,
$$PNm\left(A\right)=0.837,$$
$$PNm\left(B\right)=0.163.$$

The transformation results of different methods for Example 3 are shown in Table 2 and Figure 7.

Since the BPA of a single-subset proposition *A*, but not of a single-subset proposition *B*, directly exists in the evidence, when assigning multi-element propositions, intuitively, the proposition *A* should have a larger weight; however, in this test, the OWA method assigned a larger weight to the proposition *B*. The OVG, PraPl, PrPl, and BetP methods assigned the BPA of multi-element proposition equally to propositions *A* and *B* . The OVGWP method and the proposed method considered the prior information of $m\left(A\right)=0.5$, thus considering the connection between multi-element propositions and each single-element proposition, and assigned a larger weight to the single-subset proposition *A* . The transformation results are reasonable, and the PIC is relatively larger, which is more favorable for decision-making.

**Example** **4**.
*The frame of discernment is $\Theta =\left\{A,B,C\right\}$, and m is a mass function in Θ; then, the BPA is given by:*

$$m\left(A\right)=0.1,m\left(A,B\right)=0.2,m\left(B,C\right)=0.3,m\left(A,B,C\right)=0.4$$



Step 1: The proposition cardinality defines $m\left(A,B,C\right)=0.4$ as the input layer of the neural network; the hidden-layer proposition subsets are denoted by $\left\{A,B\right\}$, $\left\{A,C\right\}$, and $\left\{B,C\right\}$; $m\left(A\right)=0.1$ and $m\left(B,C\right)=0.3$ are bias units; $PNm\left(A\right)$, $PNm\left(B\right)$, and $PNm\left(C\right)$ denote the output layer.

Step 2: Calculate the AIC and IIC values of propositions $\left\{A,B\right\}$, $\left\{A,C\right\}$, and $\left\{B,C\right\}$ by Equations (Equation 30) and (Equation 31) as follows:$$AIC\left(A,B\right)=181.2722,$$
$$AIC\left(A,C\right)=81.4509,$$
$$AIC\left(B,C\right)=121.5104,$$
$$IIC\left(A,B\right)=270.4264,$$
$$IIC\left(A,C\right)=1339.4308,$$
$$IIC\left(B,C\right)=121.5104.$$

The weights of each proposition are obtained by Equation (Equation 32) as follows:$$W\left(A,B\right)=r\times 181.2722^{1-r}+\left(1-r\right)\times 270.4264^{r},$$
$$W\left(A,C\right)=r\times 81.4509^{1-r}+\left(1-r\right)\times 1339.4308^{r},$$
$$W\left(B,C\right)=r\times 121.5104^{1-r}+\left(1-r\right)\times 121.5104^{r}.$$

The BPAs of proposition subsets are obtained by Equations (Equation 34) and (Equation 35) as follows:$$Nm^{{}^{\prime }}\left(A,B\right)=0.2+\frac{W\left(A,B\right)}{W\left(A,B\right)+W\left(A,C\right)+W\left(B,C\right)}\times 0.4=g_{AB}\left(r\right),$$
$$Nm\left(A,B\right)=\max\left(0,g_{AB}\left(r\right)\right),$$
$$Nm^{{}^{\prime }}\left(A,C\right)=\frac{W\left(A,C\right)}{W\left(A,B\right)+W\left(A,C\right)+W\left(B,C\right)}\times 0.4=g_{AC}\left(r\right),$$
$$Nm\left(A,C\right)=\max\left(0,g_{AC}\left(r\right)\right),$$
$$Nm^{{}^{\prime }}\left(B,C\right)=0.3+\frac{W\left(B,C\right)}{W\left(A,B\right)+W\left(A,C\right)+W\left(B,C\right)}\times 0.4=g_{BC}\left(r\right),$$
$$Nm\left(B,C\right)=\max\left(0,g_{BC}\left(r\right)\right).$$

The AIC and IIC values of each single-element proposition are calculated by Equations (Equation 30) and (Equation 31) as follows:$$AIC\left(A\right)=e^{4\times \left(0.1+g_{AB}\left(r\right)+g_{AC}\left(r\right)\right)},$$
$$AIC\left(B\right)=e^{4\times \left(g_{AB}\left(r\right)+g_{BC}\left(r\right)\right)},$$
$$AIC\left(C\right)=e^{4\times \left(g_{AC}\left(r\right)+g_{BC}\left(r\right)\right)},$$
$$IIC\left(A\right)=e^{8\times \left(g_{AB}\left(r\right)+g_{AC}\left(r\right)\right)},$$
$$IIC\left(B\right)=e^{8\times \left(g_{AB}\left(r\right)+g_{BC}\left(r\right)\right)},$$
$$IIC\left(C\right)=e^{8\times \left(g_{AC}\left(r\right)+g_{BC}\left(r\right)\right)}.$$

The weights of each single-element proposition are obtained by Equation (Equation 32) as follows:$$W\left(A\right)=r\times AIC{\left(A\right)}^{1-r}+\left(1-r\right)\times IIC{\left(A\right)}^{r}=q_{A}\left(r\right),$$
$$W\left(B\right)=r\times AIC{\left(B\right)}^{1-r}+\left(1-r\right)\times IIC{\left(B\right)}^{r}=q_{B}\left(r\right),$$
$$W\left(C\right)=r\times AIC{\left(C\right)}^{1-r}+\left(1-r\right)\times IIC{\left(C\right)}^{r}=q_{C}\left(r\right).$$

The probabilities of each proposition are calculated by Equations (Equation 37) and (Equation 38):$$PNm^{{}^{\prime }}\left(A\right)=0.1+\frac{q_{A}\left(r\right)}{q_{A}\left(r\right)+q_{B}\left(r\right)}\times g_{AB}\left(r\right)+\frac{q_{A}\left(r\right)}{q_{A}\left(r\right)+q_{C}\left(r\right)}\times g_{AC}\left(r\right)$$
$$PNm\left(A\right)=\max\left(0,PNm^{{}^{\prime }}\left(A\right)\right),$$
$$PNm^{{}^{\prime }}\left(B\right)=\frac{q_{B}\left(r\right)}{q_{A}\left(r\right)+q_{B}\left(r\right)}\times g_{AB}\left(r\right)+\frac{q_{B}\left(r\right)}{q_{B}\left(r\right)+q_{C}\left(r\right)}\times g_{BC}\left(r\right)$$
$$PNm\left(B\right)=\max\left(0,PNm^{{}^{\prime }}\left(B\right)\right),$$
$$PNm^{{}^{\prime }}\left(C\right)=\frac{q_{C}\left(r\right)}{q_{A}\left(r\right)+q_{C}\left(r\right)}\times g_{AC}\left(r\right)+\frac{q_{C}\left(r\right)}{q_{B}\left(r\right)+q_{C}\left(r\right)}\times g_{BC}\left(r\right)$$
$$PNm\left(C\right)=\max\left(0,PNm^{{}^{\prime }}\left(C\right)\right).$$

When the value of parameter *r* varies in the range of [0, 1], the probability and PIC curves change, as shown in Figure 8.

Step 3: Determine the optimal probability by Equation (Equation 39):r=0.16,
$$PNm\left(A\right)=0.305,$$
$$PNm\left(B\right)=0.494,$$
$$PNm\left(C\right)=0.201.$$

The transformation results of different methods are shown in Table 3 and Figure 9.

The PrBel method is limited and cannot obtain the correct transformation results. The PraPl, ITP, and OWA methods assign a larger weight to proposition *A* than to proposition *B*, due to the direct presence of the BPA of a single-element proposition *A* in the evidence. However, it is not considered that the multi-element propositions contain single-element proposition *B* when $Pl\left(B\right)>Pl\left(A\right)$. Intuitively, a single-element proposition *B* should have a larger weight. The transformation results of the PrPl, ITP, MPSV, OVGP, and OVGPWP methods and the proposed method are reasonable. However, compared with the other methods, the proposed method has the largest PIC value of 0.0596, lower information uncertainty after transformation, and is more beneficial to the decision-making process.

## 6. Practical Application

In this section, the newly proposed method is applied to the practical problem of target recognition to further verify its effectiveness.

**Example** **5**.
*The Iris dataset can be divided into three categories :$\left\{Setosa,Versicolor,Virginica\right\}$ as FOD $\Theta =\left\{Se,Ve,Vi\right\}$, each category contains four attributes: $\left\{SL,SW,PL,PW\right\}$; then, the Iris objects are assigned as BPAs $\left\{m_{SL},m_{SW},m_{PL},m_{PW}\right\}$ in light of the attributes given in Table 4.*


The specific steps can be described as follows:

Step 1: The degree of credibility between evidence is measured according to the literature [42].

Step 2: The credibility degree of each evidence is modified.

Step 3: The weighted average evidence is obtained by taking into account the relationship between the bodies of evidence and the relative importance of the collected evidence.

Step 4: According to the Dempster combination rule, the combination result is obtained by combining the weighted average evidence three times.

Step 5: Due to the large information uncertainty in the combination result, the combination result is transformed into probability distribution by the proposed method.

For the target recognition problem in the Iris dataset, the proposed method is compared with some existing methods [35,42,43,44], as shown in Table 5.

The recognition results of the five methods are the same, and the category of Iris is identified as Ve. It can be seen from Table 5 that this method outperformed the comparison methods, and the belief in Ve is as high as 92.01%, the belief degrees of Xiao’s method [43], Jiang’s method [44], MSDF [42], MPSV [35] for the category Ve are 73.90%, 87.98%, 91.63%, and 91.86%, respectively.

## 7. Conslusions

How to reasonably transform the BPA under the DS evidence theory into probability before decision-making has become a major research hotspot. In this paper, a probability transformation method is proposed by combining the BPA with a neural network. The BPA of multi-element propositions with maximum cardinality is used as the input network layer, and the BPAs of the remaining propositions are used as bias nodes, which are assigned to the proposition subsets in each hidden layer of the neural network according to the weights. The probability of each single-element proposition is obtained as the network output. The AIC and IIC values of each proposition subset are determined using the belief and plausibility functions, respectively, and then combined to obtain the weight factors of the contained variables to output the probability containing variables. Finally, the PIC is maximized as a constraint to determine the variables to obtain the optimal probability transformation result. The proposed method is verified by numerical examples and compared with the other methods. The results indicate that the proposed method is more reasonable and has better generalizability and lower information uncertainty regarding the transformed results than the other method, which makes it more beneficial for decision-making. However, the proposed method may generate a large computational effort when the cardinality of a multi-element proposition is too large. In the future, a more comprehensive evaluation index for probabilistic transformation results could be explored to further verify the rationality and superiority of this method and apply this method to more practical scenarios.

## Figures and Tables

**Figure 1 entropy-24-01638-f001:**
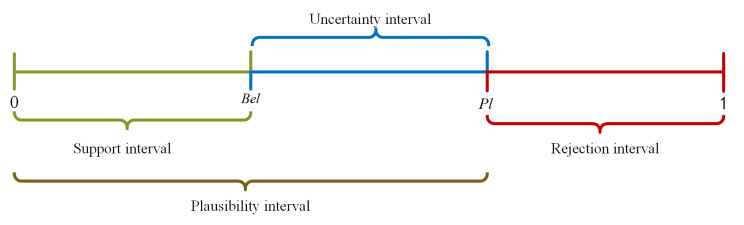
Illustration of evidence intervals of the DS evidence theory.

**Figure 2 entropy-24-01638-f002:**
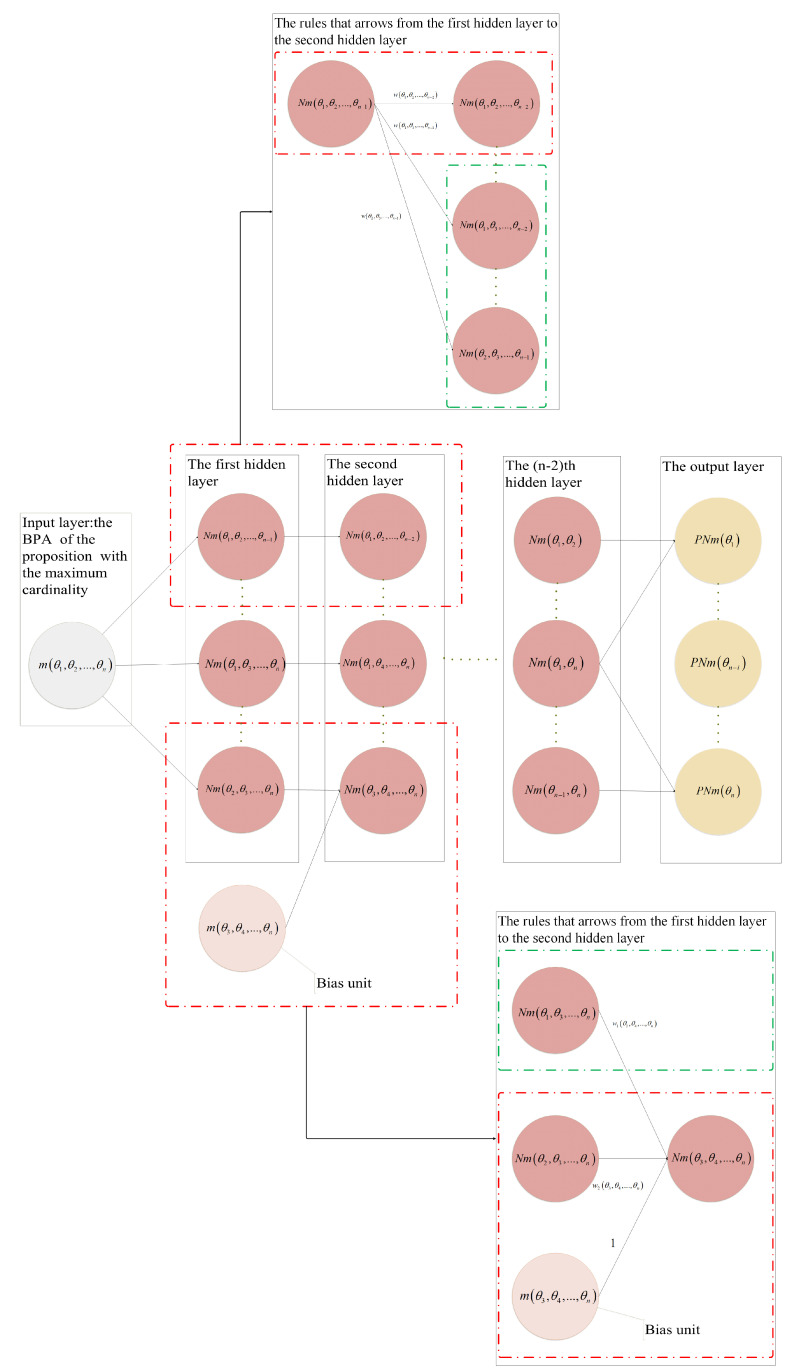
Schematic representation of the neural network structure.

**Figure 3 entropy-24-01638-f003:**
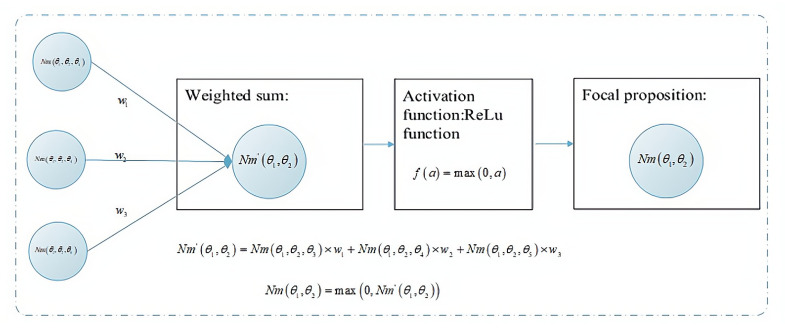
Internal structure of neurons.

**Figure 4 entropy-24-01638-f004:**
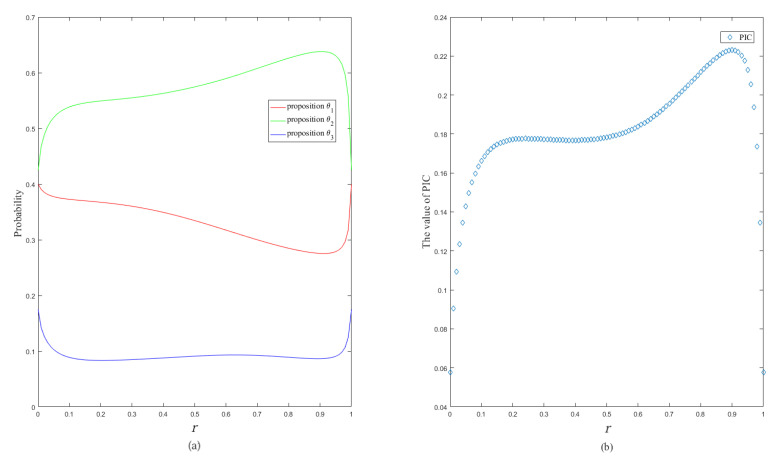
The changing trend of the single-element proposition probability (**a**) and the changing trend of PIC value (**b**).

**Figure 5 entropy-24-01638-f005:**
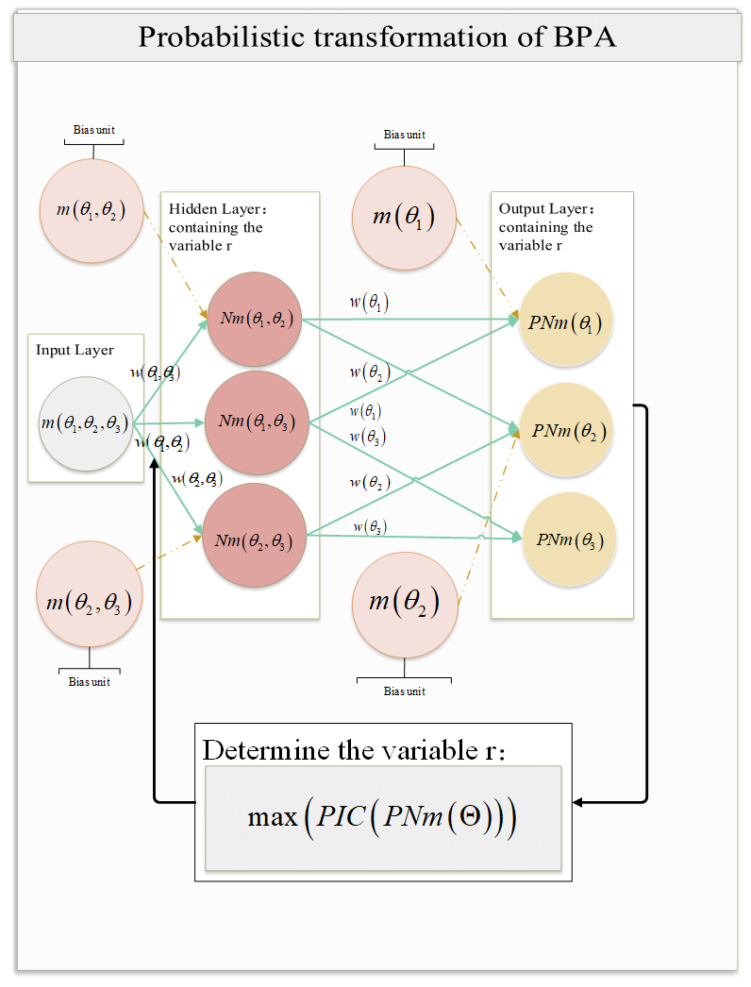
The implementation process of Example 1.

**Figure 6 entropy-24-01638-f006:**
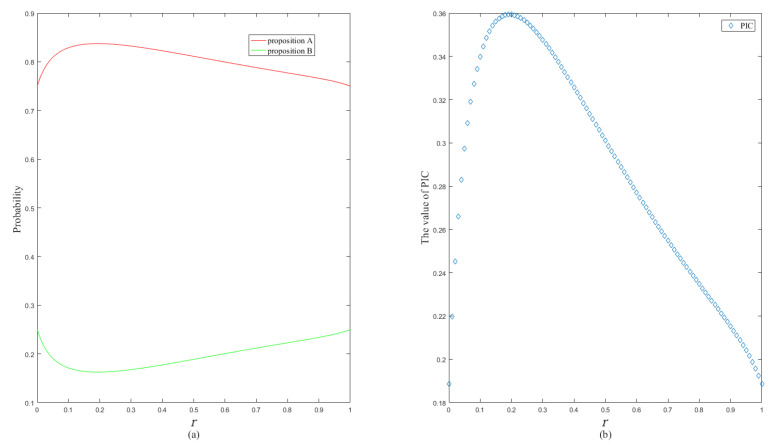
The changing trend of the single-element proposition probability (**a**) and the changing trend of PIC value (**b**).

**Figure 7 entropy-24-01638-f007:**
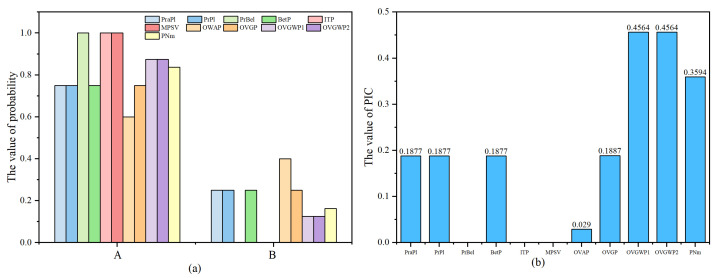
The probabilistic transformation results (**a**) and the PIC values (**b**) of different methods.

**Figure 8 entropy-24-01638-f008:**
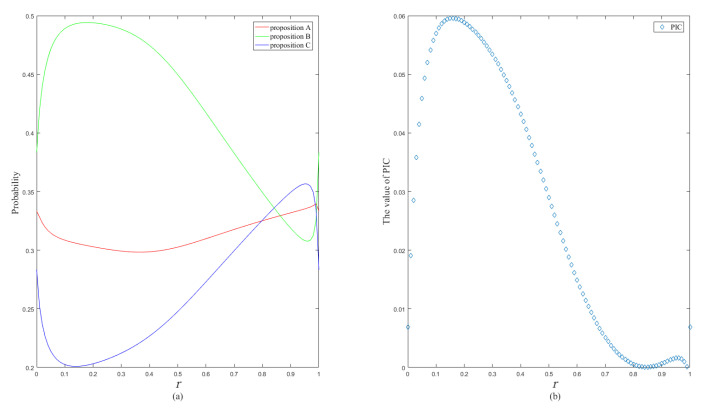
The changing trend of the single-element proposition probability (**a**) and the changing trend of PIC value (**b**).

**Figure 9 entropy-24-01638-f009:**
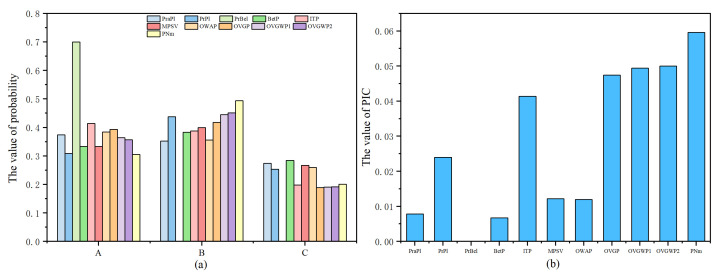
The probabilistic transformation results (**a**) and the PIC values (**b**) of different methods for Example 4.

**Table 1 entropy-24-01638-t001:** Probability transformation results of different methods for Example 2.

Method	*A*	*B*	*C*
PraPl [31]	0.2	0.4	0.4
PrPl [31]	0.2	0.4	0.4
PrBel [31]	0.2	NaN	NaN
BetP [32]	0.2	0.4	0.4
ITP [34]	0.2	0.4	0.4
MPSV [35]	0.2	0.4	0.4
OWAP [33]	0.3230	0.3756	0.3014
OVGP [36]	0.2	0.4	0.4
OVGWP1 [37]	0.2	0.4	0.4
OVGWP2 [37]	0.2	0.4	0.4
PNm	0.2	0.4	0.4

**Table 2 entropy-24-01638-t002:** Probability transformation results of different methods for Example 3.

Method	*A*	*B*	PIC
PraPl [31]	0.75	0.25	0.1877
PrPl [31]	0.75	0.25	0.1877
PrBel [31]	1	NaN	NaN
BetP [32]	0.75	0.25	0.1877
ITP [34]	1	0	NaN
MPSV [35]	1	0	NaN
OWAP [33]	0.6	0.4	0.0290
OVGP [36]	0.75	0.25	0.1887
OVGWP1 [37]	0.875	0.125	0.4564
OVGWP2 [37]	0.875	0.125	0.4564
PNm	0.837	0.163	0.3594

**Table 3 entropy-24-01638-t003:** Probability transformation results of different methods for Example 4.

Method	*A*	*B*	*C*	PIC
PraPl [31]	0.374	0.352	0.274	0.0078
PrPl [31]	0.309	0.438	0.253	0.0240
PrBel [31]	0.7	NaN	NaN	NaN
BetP [32]	0.333	0.383	0.284	0.0067
ITP [34]	0.414	0.388	0.198	0.0414
MPSV [35]	0.333	0.400	0.267	0.0122
OWAP [33]	0.384	0.356	0.260	0.0120
OVGP [36]	0.393	0.418	0.189	0.0474
OVGWP1 [37]	0.364	0.445	0.191	0.0494
OVGWP2 [37]	0.357	0.451	0.192	0.0500
PNm	0.305	0.494	0.201	0.0596

**Table 4 entropy-24-01638-t004:** Modeled BPA of an object from Iris dataset.

BPA	$m_{\mathrm{SL}}$	$m_{\mathrm{SW}}$	$m_{\mathrm{PL}}$	$m_{\mathrm{PW}}$
$m\left(Se\right)$	0.0437	0.0865	$1.40\times 10^{-9}$	$8.20\times 10^{-6}$
$m\left(Ve\right)$	0.3346	0.2879	0.6570	0.6616
$m\left(Vi\right)$	0.2916	0.1839	0.1726	0.1692
$m\left(Se,Ve\right)$	0.0437	0.0863	$1.30\times 10^{-9}$	$8.20\times 10^{-6}$
$m\left(Se,Vi\right)$	0.0239	0.0865	$1.40\times 10^{-11}$	$3.80\times 10^{-6}$
$m\left(Ve,Vi\right)$	0.2385	0.1825	0.1704	0.1692
$m\left(Se,Ve,Vi\right)$	0.0239	0.0863	$1.40\times 10^{-11}$	$3.80\times 10^{-6}$

**Table 5 entropy-24-01638-t005:** Results of the target recognition problem in Iris dataset.

Method	$m\left(\mathrm{Se}\right)$	$m\left(\mathrm{Ve}\right)$	$m\left(\mathrm{Vi}\right)$	Target
Xiao’s method [43]	0.0053	0.7390	0.2407	Ve
Jiang’s method [44]	$4.90\times 10^{-4}$	0.8798	0.1130	Ve
MSDF [42]	$6.88\times 10^{-5}$	0.9163	0.0790	Ve
MPSV [35]	$7.24\times 10^{-5}$	0.9186	0.0813	Ve
PNm	0.0001	0.9201	0.0798	Ve

## Data Availability

Not applicable.

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
