# Peer review of "A Decision Probability Transformation Method Based on the Neural Network"

_entropy, 2022, doi:10.3390/e24111638_

Round 1

Reviewer 1 Report

This article proposes a probability transformation method based on neural network to achieve the transformation from the BPA to the probabilistic decision. However, there are still some areas for improvement in the article:

1.     The abstract part lacks the description of the innovative points of the model. The authors are suggested to explain the out-performance of the proposed method.

2.     In the first paragraph of the Introduction Section, the literature is redundant, please highlight the focus of the research area. Besides, references related to neural networks should be introduced to show the current research status.

3.     In page 2, ‘Deng [34] proposed a probability……probability of single-element proposition.’ and ‘The main question considered in this study is……proposition containing the parameter is output.’ should be revise the syntax to make the text more logical.

4.     In page 2, some sentences are vague and confusing the reader, including ‘Although this method does not produce counterintuitive conversion results……’ and ‘As varies, the weighting function reaches its extreme value……’. The authors need to specify the demonstrative pronoun.

5.     In page 2, ‘Then, the probabilistic information content (PIC) is determined…….’. Please bring in the necessary references.

6.     In Introduction Section, the authors should point out the shortcomings of existing methods and the gaps that this article fills.

7.     In addition, some current related works of entropy and decision-making are suggested to discuss, e.g., Negation of the quantum mass function for multisource quantum information fusion with its application to pattern classification; Generalized divergence-based decision making method with an application to pattern classification; A complex weighted discounting multisource information fusion with its application in pattern classification.

8.     In Section 2.2, the authors introduced several well-known methods. However, the key points of each works are not obvious. Please introduce the innovation points of these works.

9.     At the beginning of the Section 4. The authors pointed ‘To overcome the shortcomings of the existing methods……’. Please introduce the shortcomings specifically.

10.  In method section, the authors said ‘the BPA  of the proposition  with the maximum cardinality is used as the input network layer.’. However, what if there is more than one BPA of the proposition with maximum cardinality?

11.   In Fig.2, the rules that arrows from the first hidden layer to the second hidden layer should be explained.

12.   Please improve the quality of Fig 3 and explain how the Relu function works.

13.   In Section 5, the authors are suggested to apply the model to real data to demonstrate the performance of the proposed method.

14.   In Conclusion Section, it is suggested to analyze the limitations and shortcomings of this work, and point out the future research direction.

In short, the work is valuable but still needs some revisions before its acceptance.

Author Response

Dear reviewer, thank you for your guidance and suggestions for my paper, which is helpful for my paper 's promotion. I will add an attachment to reply to each of your suggestions in detail.

Reviewer 2 Report

This paper focuses on the probability transformation of basic probability assignment, which is an important stage for decision making in the framework of Dempster-Shafer evidence theory. A neural network-based method is proposed to realize the transformation. The rationality and effectiveness of the proposed method is demonstrated via several examples. This paper is well organized and easy to read. However, I have several critical concerns about the methodology and experiments.

1. The necessity of using neural network to realize the probability transformation is unconvincing. Generally, the neural network is used to fit a model by learning the model parameters (weights and bias). But in the proposed method, the weights and bias of the neural network are known. As shown in the main operations of the method in Page 12, there is no learning process and the probability transformation can be realized by several deterministic equations. Therefore, the proposed method does not take advantage of the learning ability of the neural network.

2. In Line 157, you mentioned “To overcome the shortcomings of the existing methods,…”. But, the shortcomings of the existing methods were not well analyzed. So, the motivation of the proposed method is not clear.

3. Eqs. (30) and (31) give the calculations of AIC and IIC directly without explanations . Are they new definitions or referred from other’s work? You should justify it.

4. In Pages 8 and 9, there are two proofs. But there are no theorems above them.

5. In both the methodology and experiments, you use the PIC to evaluate the performance of different probability transformation methods. Is this performance measure widely accepted? Or is there any other performance measure?

6 What are the meanings of the longitudinal coordinates of the left figures in Figs. 7 and 9?

7. As shown in Fig. 7, the proposed method did not achieve the best performance. Please explain it.

8. Only several toy examples are used to evaluate the proposed method. It is recommended to use more complex data with real application background.

Author Response

(The authors gave the same response as above.)

Reviewer 3 Report

See the attachment.

Author Response

(The authors gave the same response as above.)

Round 2

Reviewer 2 Report

Most of my concerns have been addressed. The revised version seems better. But I still have two concerns:

1. The example of Iris data given in Section 6 is not a practical problem of target recognition. It is just an example of data classification.

2. The linguistic quality needs improvement. It is essential to make sure that the manuscript reads smoothly. Consult a professional or use a language editing service.

Author Response

Dear reviewer, I have made a detailed reply to your comments in the annex.
